# A Case Series and Literature Review of Telogen Effluvium and Alopecia Universalis after the Administration of a Heterologous COVID-19 Vaccine Scheme

**DOI:** 10.3390/vaccines11020444

**Published:** 2023-02-15

**Authors:** Jenny Hernández Arroyo, Juan S. Izquierdo-Condoy, Esteban Ortiz-Prado

**Affiliations:** 1Centro Médico Capilar, Quito 170517, Ecuador; 2One Health Research Group, Faculty of Medicine, Universidad de las Américas, Quito 170137, Ecuador

**Keywords:** SARS-CoV-2, COVID-19, vaccines, adverse events, alopecia areata

## Abstract

COVID-19 vaccines have positively changed the course of the pandemic. They entered the market after only one year of the initial trials, which that yielded positive results in terms of safety and efficacy. However, after inoculating billions of people in the most extensive vaccination campaign worldwide, mild but common and some rare but potentially fatal adverse events have been reported. Among several self-reported adverse events, hair loss and alopecia have been linked to COVID-19 mRNA or viral vector vaccines. We tracked and followed a series of five cases with post-vaccine telogen effluvium and alopecia development in Ecuador. Here, we reported the clinical presentation of two women and three men with the diagnosis of post-vaccine hair loss. All patients received a heterologous vaccination scheme (mRNA and attenuated virus vaccine) with an additional viral vector booster associated with the apparition of telogen effluvium and alopecia universalis between 3 and 17 days after the vaccine was administered.

## 1. Introduction

The development of vaccines throughout human history has changed the course of diseases and humanity [1]. The process of vaccine development often requires decades to obtain a marketable vaccine; nevertheless, with the advent of the SARS-CoV-2 pandemic in March 2020, the process was accelerated by both necessity and the amount of resources available for this purpose [2]. While this process has followed the highest standards of safety and control, due to the large number of people vaccinated in all corners of the globe, reports of adverse events are still very common [3,4]. In this context, by December 2020, the Food and Drug Administration (FDA) authorized the emerging use of the mRNA-based vaccine produced by Pfizer/BioNTech (BNT162b2), followed by Moderna (ARNm-1273), and later the rest of the COVID-19 vaccines, including the viral-vector-based AstraZeneca (ChAdOx1), or the attenuated and inactivated virus vaccine from Sinopharm (BBIBP-CorV) and Sinovac (Sinovac-CoronaVac) [5]. With initial reports on safety and effectivity, and after the biggest vaccination campaign took place all over the planet, adverse events started to appear [5,6]. Among the most common and expected are pain at the injection site, fever, headache, and malaise, as well as more rare events, such as transverse myelitis, Guillen–Barre syndrome, or platelet activation by vaccine-induced immune thrombocytopenia [4,7,8].

Regardless of the type of COVID-19 vaccine or the wide range of adverse events that have been described within the Vaccine Adverse Event Reporting System (VAERS), some rare and poor study events, such as menstrual rhythm alteration, myocarditis, and alopecia, have been exceptionally described [3,9,10,11]. From these, alopecia areata and hair loss, the two common symptoms described by men and women around the world, have caught our attention, especially after the first report was described [12,13].

Alopecia refers to abnormal hair loss that occurs because of a pathological process. In non-scarring cases, there is an exaggerated entry of hair into the telogen stage without affecting the integrity of the hair follicle, making it potentially reversible [14]. Although the exact mechanism of alopecia areata has not been described to date, the pathophysiological basis of hair loss is considered to be constituted on the breakdown of the immune privilege of the follicle, which is characterized by an increase in the number of NK lymphocytes in the affected follicles dependent on the JAK–STAT pathway. In addition, it is considered that the appearance of alopecia areata has an important genetic influence. Between 10% and 20% of those affected have relatives with the condition, and the genes that are affected in these scenarios are associated with alterations in the immune and inflammatory response genes, such as MHC, CTLA4, and PRDX5 [15]. Furthermore, the influence of several external factors has been postulated, such as exposure to emotional stress, use of drugs, viral infections (cytomegalovirus, Epstein–Barr virus, and influenza virus), and certain vaccines [15,16].

In some countries, regulatory agencies have published data on alopecia areata as an adverse event following COVID-19 vaccination. In the United States, the Center for Disease Control and Prevention (CDC) reported a total of 114 cases of alopecia areata, 1 case of alopecia totalis, and 1 case of alopecia universalis, following the administration of the Pfizer/BioNTech (66%) and Moderna (29%) vaccines [17]. In the United Kingdom, 154 cases of alopecia areata were reported, mainly caused by the Pfizer/BioNTech (50%) and AstraZeneca (40%) vaccines [18]. However, so far, no cases of alopecia areata following the administration of COVID-19 vaccines have been reported in the Latin American region, where most of the population has different phenotypic characteristics and received different vaccination schedules compared to those of developed countries; therefore, the inflammatory response that accompanies the administration of COVID-19 vaccines may present different features in Latin American populations and trigger adverse events, such as hair loss.

For these reasons, a report of five cases of individuals with hair loss following the administration of COVID-19 vaccines and a review of the literature on this phenomenon were carried out.

## 2. Cases Presentation

### 2.1. Case 1

Case 1 concerned a female patient who was 27 years old and had no family history of alopecia areata. She also had no personal history of COVID-19, but she presented a personal history of polycystic ovary syndrome and alopecia universalis 6 years prior. She presented for a medical consultation on 2 February 2022, claiming to have noticed excessive hair loss 8 days after receiving the first booster of the AstraZeneca vaccine. After a physical examination, she was diagnosed with alopecia universalis (Figure 1(A.1,A.2)). In laboratory tests, hematological and chemical parameters did not show abnormal values.

### 2.2. Case 2

Case 2 concerned a female patient who was 51 years old and had a family history of hypothyroidism (maternal grandmother). She had no personal history of COVID-19, but she had a personal history of hypertension. She attended medical consultation on 24 August 2022, reporting a large amount of hair loss 3 days after receiving the first booster of the AstraZeneca vaccine. After a physical examination, telogen effluvium was diagnosed (Figure 1(B.1,B.2)). In laboratory tests, hematological and chemical parameters did not show abnormal values.

### 2.3. Case 3 

Case 3 concerned a male patient who was 34 years old and had a family history of type 2 diabetes mellitus (maternal grandmother) and arterial hypertension (maternal grandmother). He presented a history of COVID-19 after contracting it once and no personal history of alopecia. He attended a medical consultation on 26 February 2022, reporting a large amount of hair loss 10 days after receiving the first booster of the AstraZeneca vaccine. After a physical examination, telogen effluvium was diagnosed (Figure 1(C.1,C.2)). In the laboratory tests, no hematological alterations were found, while in the chemistry tests, the only alteration evidenced was elevated creatinine (1.31 mg/dL).

### 2.4. Case 4

Case 4 concerned a male patient who was 40 years old and had no family history of alopecia. He presented a history of COVID-19 after contracting it once and no personal history of alopecia. He attended a medical consultation on 26 February 2022, reporting a large amount of hair loss 7 days after receiving the first booster of the AstraZeneca vaccine. After a physical examination, telogen effluvium was diagnosed (Figure 1(D.1,D.2)). In laboratory tests, hematological and chemical parameters did not show abnormal values.

### 2.5. Case 5

Case 5 concerned a male patient who was 59 years old and had no family history of alopecia areata, although he did present a personal history of hypertension and COVID-19 infection after contracting it once. He presented for a medical consultation on 21 April 2022, claiming to have noticed excessive hair loss 17 days after receiving the first booster of the AstraZeneca vaccine. After a physical examination, he was diagnosed with alopecia universalis (Figure 1(E.1–E.3)). In the laboratory analysis, no hematologic alterations were found. In relation to lipids, total cholesterol (256.6 mg/dL) and triglyceride (313 mg/dL) values were increased. In addition, ferritin value was increased (1415 ng/mL).

In all the cases reported, the base vaccination schedule (first dose and second dose) received by the patients was a mixed vaccination schedule, characterized by an initial dose of the Pfizer/BioNTech vaccine, followed by a second dose with the Sinovac vaccine. Likewise, in all the cases reported, mesotherapy measures were applied, achieving high percentages of rapid improvement in the cases that had a diagnosis of telogen effluvium; meanwhile, in the cases of alopecia universalis, improvement was much lower, as the usual late response to this pathology can take 1 year or more to be controlled, rather than cured. The details of the patients studied are reported in Table 1.

All patients gave their approval for the publication of clinical information and medical images through a personal informed consent document.

## 3. Discussion

As of November 1, 2022, a total of 7 reports have been published that include 19 individuals in which alopecia was associated with COVID-19 vaccines. The available resources are limited to case reports and case series that have exposed the development of alopecia areata (n = 15), alopecia totalis (n = 1), and alopecia universalis (n = 3) [11,12,19,20,21,22,23]. According to sex, the available data show that 68.4% (n = 13) corresponded to women; meanwhile, among our participants, the most cases were male.

On the other hand, despite the known genetic influence on the development of alopecia areata, only one case reported in the literature claimed to have a family history of alopecia areata [20], and none of our participants had a family history of alopecia areata.

Regarding personal histories of alopecia areata, 52.6% (n = 10) of reports claimed that participants had a personal history of alopecia areata [11,12,19,20,21]; however, within our participants, only one woman (20%) recorded a personal history of alopecia universalis prior to the administration of COVID-19 vaccines.

Among the vaccines attributed to the development of alopecia areata, the Pfizer/BioNTech vaccine is the most reported (52.6% (n = 10) of cases), followed by the AstraZeneca vaccine (31.6% (n = 6)), and Moderna (15.8% (n = 3)). In all cases, homogeneous vaccination schedules were administered. Likewise, according to the number of doses, apparently alopecia areata occurred mostly among those who received a second dose of vaccination 57.9% (n = 11); meanwhile, among our patients, 100% of adverse events occurred after the first booster (third dose). Detailed information on the findings collected is summarized in Appendix A.

Although the relationship between the use of vaccines and the development of alopecia areata has been proposed for several years and has been attributed to different vaccines, such as the Hepatitis B vaccine and the poliovirus vaccine, until now, it has been impossible to describe the exact mechanism of this relationship [24]. However, it seems that several of the vaccines against the SARS-CoV-2 virus are potential triggers of hair loss. We consider that the phenotypic differences of our participants, who are of mixed ethnicity, may explain the development of telogen effluvium and alopecia despite a predominant absence of a family and personal history of alopecia. This proposal can be supported by the findings of a large study conducted in the United States, where Hispanic women had a higher risk of alopecia areata compared to White women [25], a racial disparity that could be explained using an immunological approach because autoimmune diseases have been shown to attack Hispanic and Black people more severely [26]. TNF alpha is mainly responsible for this increased immune activity and has also been found to be increased in inflammatory lesions of alopecia areata [27,28].

After an exhaustive literature search, it was possible to consider that this is the first report in the world to consider the development of telogen effluvium and alopecia areata in individuals who received mixed vaccination schedules, characterized using the Pfizer and Sinovac vaccines, and presumably triggered by the administration of the first booster with the AstraZeneca vaccine. Furthermore, according to the available reports in the literature, 52.6% of those who developed alopecia areata following COVID-19 vaccination had a personal history of alopecia, which could mean that, in these patients, the vaccines simply aggravate the individual’s previous condition, whereas in our patients, only one of them (20%) had a personal history of alopecia, and the hair loss developed within a short period (less than 17 days) after the administration of the first booster of the COVID-19 vaccine which, in all cases, was the AstraZeneca vaccine. In Ecuador, only one study has so far explored adverse events following vaccination against COVID-19, and among 25 symptoms evaluated, there was no report of hair loss or alopecia areata [8], making this the first report in the Ecuadorian population. All these findings could be interpreted as a possible adverse effect triggered by the administration of mixed vaccination schedules; however, larger studies with more controlled methodologies will be necessary to propose a causal hypothesis. 

Although the present study supports the hypothesis that there may be some association between hair loss and COVID-19 vaccines and adds previously undescribed information, such as hair loss in users of mixed vaccination schedules and the Sinovac vaccine, our data do not provide sufficient support to advise against the use of COVID-19 vaccines. Furthermore, as demonstrated, along with the 19 cases previously reported in the literature, hair loss events from COVID-19 vaccines are extremely rare, and several factors appear to be involved with their development. We believe that the reported risk of alopecia due to COVID-19 vaccines is outweighed by the benefits, such as decreased severity and mortality, as well as the probable effect of decreasing the long-term sequelae of infection (long-COVID) [29], a condition that has already been described among Ecuadorians [30,31].

## 4. Conclusions

This research supports the hypothesis that there is some association between hair loss and COVID-19 vaccination and expands on the theoretical rationale for the use of mixed vaccination schedules and the Sinovac vaccine. As demonstrated, along with the 19 cases previously reported in the literature, hair loss events from COVID-19 vaccines are extremely rare but should be recognized as potential adverse events attributable to COVID-19 vaccines for public awareness. Furthermore, the benefits of COVID-19 vaccines considerably outweigh the reported risk of hair loss and alopecia areata.

## Figures and Tables

**Figure 1 vaccines-11-00444-f001:**
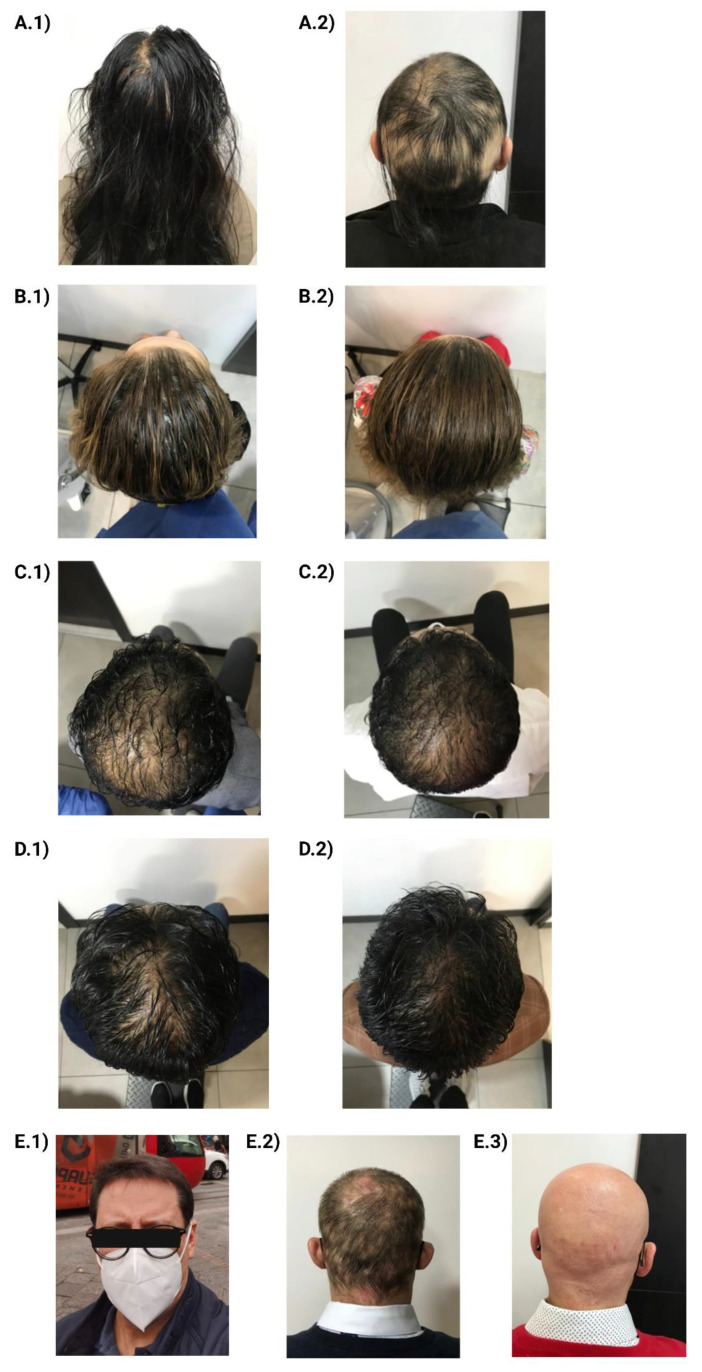
Female patient (27 years old) with alopecia universalis post-AstraZeneca vaccine; (**A.1**) photographic record prior to vaccination; (**A.2**) alopecia universalis after COVID-19 vaccination. Female patient (51 years old) with telogen effluvium pos- AstraZeneca vaccination; (**B.1**) photographic record of telogen effluvium at medical consultation after administration of the vaccine; (**B.2**) photographic record of evolution after treatment (80% improvement). Male patient (34 years old) with telogen effluvium after AstraZeneca vaccine; (**C.1**) photographic record of telogen effluvium at medical consultation after administration of the vaccine; (**C.2**) photographic record of evolution after treatment (50% improvement). Male patient (40 years old) with telogen effluvium after AstraZeneca vaccine; (**D.1**) photographic record of telogen effluvium at medical consultation after administration of the vaccine; (**D.2**) photographic record of evolution after treatment (90% improvement). Male patient (59 years old) with alopecia universalis post-AstraZeneca vaccine; (**E.1**) personal photographic record prior to vaccine administration; (**E.2**) alopecia areata post-vaccination; (**E.3**) alopecia universalis after treatment.

**Table 1 vaccines-11-00444-t001:** Case characteristics of patients with hair loss after COVID-19 vaccination.

N.	Sex	Age (Years)	Ethnicity	Family History	Personal History	Vaccine Received	Symptoms	Post-Vaccination Onset Time	Therapy	Evolution
1	F	27	Mestizo	No	AU, Polycystic ovary	Pfizer, SinoVac, and AstraZeneca (Booster)	Alopecia universalis	8 days	Mesotherapy and pulses of dexamethasone and Clobetasol propionate 0.5% topical	25%
2	F	51	Mestizo	Hypothyroidism (grandmother)	AH	Pfizer, SinoVac, and AstraZeneca (Booster)	Telogen effluvium	3 days	Mesotherapy	80%
3	M	34	Mestizo	DM 2, AH (grandmother)	COVID-19	Pfizer, SinoVac, and AstraZeneca (Booster)	Telogen effluvium	10 days	Mesotherapy	50%
4	M	40	Mestizo	No	COVID-19	Pfizer, SinoVac, and AstraZeneca (Booster)	Telogen effluvium	7 days	Mesotherapy	90%
5	M	59	Mestizo	No	AH, COVID-19	Pfizer, SinoVac, and AstraZeneca (Booster)	Alopecia universalis	17 days	Mesotherapy and pulses of dexamethasone and Clobetasol propionate 0.5% topical	15%

F: female; M: male; AU: alopecia universalis; DM 2: diabetes mellitus type 2; AH: arterial hypertension.

## Data Availability

Not applicable.

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
