# Peer review of "A Case Series and Literature Review of Telogen Effluvium and Alopecia Universalis after the Administration of a Heterologous COVID-19 Vaccine Scheme"

_vaccines, 2023, doi:10.3390/vaccines11020444_

Round 1

Reviewer 1 Report

The manuscript titled "A case series and literature review of Telogen effluvium and Alopecia Universalis after the administration of a heterologous COVID-19 vaccine scheme" by Arroyo et al., has been reviewed.

In this manuscript, author did not present any direct scientific evidence which can confirm that COVIF-19 vaccines are responsible for hair loss for the presented case reports

In Figure 1 E, the age of the patient is 59 in Table-1 and description, however it is mention 27 year old in figure legends.

In table-2 "Alopecia areata after SARS-CoV-2 vaccination" authors presented this paper for two different authors. 

1. Scollan M et al.

2. Gallo et al., 

Author Response

Comments and Suggestions for Authors

The manuscript titled "A case series and literature review of Telogen effluvium and Alopecia Universalis after the administration of a heterologous COVID-19 vaccine scheme" by Arroyo et al., has been reviewed.

In this manuscript, author did not present any direct scientific evidence which can confirm that COVIF-19 vaccines are responsible for hair loss for the presented case reports. 

Although the case series design does not allow us to demonstrate causal relationships, we consider that the characteristics of the patients described as the absence of a history of alopecia and the appearance of hair loss in a very short period (17 days or less) after receiving the COVID-19 vaccine booster may support the possibility of the existence of a causal relationship between the vaccines and hair loss.

In Figure 1 E, the age of the patient is 59 in Table-1 and description, however it is mention 27 year old in figure legends.

Thanks. It was a mistake, the age of patient 5 is 59 years old. It was corrected in the whole manuscript.

In table-2 "Alopecia areata after SARS-CoV-2 vaccination" authors presented this paper for two different authors. 

  1. Scollan M et al.
  2. Gallo et al., 

There was probably some confusion. The article by Scollan et al. is "Alopecia areata after SARS-CoV-2 vaccination". And the article by Gallo et al. is "Alopecia areata after COVID-19 vaccination". Manuscript information has been added correctly.

Reviewer 2 Report

The case report “A case series and literature review of Telogen effluvium and 2 Alopecia Universalis after the administration of a heterologous 3 COVID-19 vaccine scheme” present five cases of post-vaccine telogen effluvium and Alopecia development.

Although the general idea of the manuscript is of potential interest, the  authors should improve several points.

The methodology presented is not convincing and is incomplete. Would be very important to present information on additional data such as: vaccine antibody levels, data on previous SARS-Cov 2 infection, clinical chemistry, and hematology. Which was the vaccine initially applied in each case?

Editors does not consider approval necessary the approval of an ethics committee?

The authors should improve their introduction, discussion and conclusions incorporating a hypothesis about the biological implications of these observations.

Author Response

Comments and Suggestions for Authors

The case report “A case series and literature review of Telogen effluvium and 2 Alopecia Universalis after the administration of a heterologous 3 COVID-19 vaccine scheme” present five cases of post-vaccine telogen effluvium and Alopecia development.

Although the general idea of the manuscript is of potential interest, the authors should improve several points.

The methodology presented is not convincing and is incomplete. Would be very important to present information on additional data such as: vaccine antibody levels, data on previous SARS-Cov 2 infection, clinical chemistry, and hematology. Which was the vaccine initially applied in each case?

Thanks for your comments. We have added all the information requested based on the information available for each of the cases.

Editors does not consider approval necessary the approval of an ethics committee?

The authors should improve their introduction, discussion and conclusions incorporating a hypothesis about the biological implications of these observations.

Thank you for your suggestion. We address the biological implications of our findings in the manuscript.

Reviewer 3 Report

This is a short paper that presents 5 cases of alopecia in individuals following vaccination with the Pfizer COVID 19 and Sino Vac followed by booster with AstraZeneca vaccine.  The paper presents causative evidence that the alopecia in these patients is related to the COVID vaccination protocol, because the hair loss occurred within 17 days or less of the booster; only 1/5 individuals had a family history of alopecia and the patients had no issues with hair loss prior to receiving the vaccine protocol.

This paper is confirmatory of prior publications in the literature reporting alopecia as an adverse reaction to COVID vaccination.  As such the paper does not present new evidence with the exception that this is the first report of this causative effect in Ecuador and the first report of hair loss following booster vaccinations.

Authors you make the valid point that the adverse effect of alopecia as the result of COVID vaccines is not a severe reaction that precludes the use of the vaccine.  However, could you perhaps state this conclusion more directly by indicating that individuals should be aware that alopecia is a possible adverse reaction from this vaccination regime?  Would you suggest that alopecia be added to the list of adverse reactions on vaccine information sheets?

Other suggestions:

1. Table 2 is probably not necessary in the body of the paper.  Perhaps include this as a supplementary Table or summarize these results from prior studies in the narrative.  

2. In lines 162-163 it states "This section may be divided by subheadings. It should provide a concise and precise description of the experimental results, their interpretation, as well as the experimental conclusions that can be drawn."  Not sure if this is supposed to be in your conclusion?

Author Response

Comments and Suggestions for Authors

This is a short paper that presents 5 cases of alopecia in individuals following vaccination with the Pfizer COVID 19 and Sino Vac followed by booster with AstraZeneca vaccine.  The paper presents causative evidence that the alopecia in these patients is related to the COVID vaccination protocol, because the hair loss occurred within 17 days or less of the booster; only 1/5 individuals had a family history of alopecia, and the patients had no issues with hair loss prior to receiving the vaccine protocol.

This paper is confirmatory of prior publications in the literature reporting alopecia as an adverse reaction to COVID vaccination.  As such the paper does not present new evidence with the exception that this is the first report of this causative effect in Ecuador and the first report of hair loss following booster vaccinations.

Thanks for your comment. We believe that in addition to the above, this report adds information on an essential aspect that has not been previously reported, which is the use of mixed vaccination schedules.

Authors you make the valid point that the adverse effect of alopecia as the result of COVID vaccines is not a severe reaction that precludes the use of the vaccine.  However, could you perhaps state this conclusion more directly by indicating that individuals should be aware that alopecia is a possible adverse reaction from this vaccination regime?  Would you suggest that alopecia be added to the list of adverse reactions on vaccine information sheets?

We appreciate your comment. We evaluate the severity of alopecia as an adverse event based on whether it can represent a risk to the life of the patients, that is why we consider it as mild. We added a suggestion about the importance of adding alopecia to the list of adverse reactions.

Other suggestions:

  1. Table 2 is probably not necessary in the body of the paper.  Perhaps include this as a supplementary Table or summarize these results from prior studies in the narrative.  

Thanks for your suggestion. We will move Table 2 as a supplementary table.

  1. In lines 162-163 it states "This section may be divided by subheadings. It should provide a concise and precise description of the experimental results, their interpretation, as well as the experimental conclusions that can be drawn."  Not sure if this is supposed to be in your conclusion?

Thank you for your suggestion. This is an error in the template file. The sentence has been removed.

Round 2

Reviewer 1 Report

Authors significantly revised the manuscript and it can be accepted for publication.

Reviewer 2 Report

The manuscript has been improved according to suggestions.